# Augmented Reality-Assisted Periosteum Pedicled Flap Harvesting for Head and Neck Reconstruction: An Anatomical and Clinical Viability Study of a Galeo-Pericranial Flap

**DOI:** 10.3390/jcm9072211

**Published:** 2020-07-13

**Authors:** Salvatore Battaglia, Stefano Ratti, Lucia Manzoli, Claudio Marchetti, Laura Cercenelli, Emanuela Marcelli, Achille Tarsitano, Alessandra Ruggeri

**Affiliations:** 1Oral and Maxillo-Facial Surgery Unit, IRCCS Policlinico di Sant’Orsola, Department of Biomedical and Neuromotor Sciences (DIBINEM), University of Bologna, 40100 Bologna, Italy; salbatt@yahoo.it (S.B.); claudio.marchetti@unibo.it (C.M.); 2Human Anatomy, Department of Biomedical and Neuromotor Sciences (DIBINEM), University of Bologna, 40100 Bologna, Italy; stefano.ratti@unibo.it (S.R.); lucia.manzoli@unibo.it (L.M.); alessandra.ruggeri@unibo.it (A.R.); 3Bioengineering Lab, Department of Experimental, Diagnostic and Specialty Medicine—DIMES, University of Bologna, 40100 Bologna, Italy; laura.cercenelli@unibo.it (L.C.); emanuela.marcelli@unibo.it (E.M.)

**Keywords:** reconstruction, head and neck reconstruction, 3D technologies, augmented reality, temporalis flap, periosteum flap

## Abstract

Head and neck reconstructive surgeons have recently explored new perspectives in bone restoration using periosteum carrier flaps. Following this idea, we explored the possibility of harvesting a galeo-pericranial flap. The present work studies the vascular supply of the pericranial temporo-parietal region in order to assess the possibility of harvesting a galeo-pericranial flap based on the superficial temporalis vascularization. Anatomical dissections were performed at the Anatomical Institute of the University of Bologna on eight donor cadavers. Then we performed the harvesting of the flap in vivo on eight patients. We introduced augmented reality (AR) to facilitate anatomical visualisation during free flap harvesting. Augmented reality merges virtual and actual objects, allowing direct observation of patient anatomy and the surgical field. No post-operative major or minor complications occurred. We encountered no post-operative functional issues on the donor or recipient sites, and good clinical healing was observed in all patients. In conclusion, we believe that the galea-pericranium flap could represent a new donor site for the harvesting of a periosteum carrier flap.

## 1. Introduction

Facial contours and tissues have been corrected using a variety of surgical reconstructive procedures, including autogenous bone and cartilage grafts, fat, muscle, alloplastic materials, and microvascular free flaps [1,2,3,4]. The gold standard for both soft tissue and bony reconstruction of large head and neck surgical defects is the microvascular free flap [2,3]. Although head and neck reconstructive surgery has achieved excellent results, the shortcomings of such methods have prompted a search for improvements. 

Given the limited supply of autogenous bone and postoperative donor site morbidity, bone graft substitutes have attracted particular interest. The harvesting of microvascular bony free flaps imposes a biological cost on patients. Thus, bone restoration using periosteum flaps has recently been explored. The radial region and femoral medial condyle are the most common donor sites for free periosteal grafts for head and neck reconstruction [5,6]. However, harvesting flaps from these sites is challenging due to the small periosteal area. Thus, periosteal flaps play a secondary role in head and neck reconstruction.

We explored the possibility of harvesting a galeo-pericranial flap. Galea and temporalis fascia flaps are pedicled flaps that receive blood from the superficial temporalis vessels [1,7,8,9]. 

We studied the vascular supply to the pericranial temporoparietal region in order to determine whether a galeo-pericranial flap with superficial temporalis vascularisation could be harvested.

The temporal region is an important donor site during head and neck reconstructive surgery, from which pedicled flaps, including muscle, fascia, and skin flaps, can be harvested. [1,7,8,9]. However, the anatomical and clinical viability of composite flaps with a periosteal layer in the temporoparietal region remain unclear. New computer-assisted technologies assist head and neck surgeons in clinical practice [10,11], as does augmented reality (AR). We recently used AR to facilitate anatomical visualisation during free flap harvesting [12]. Augmented reality merges virtual and actual objects, allowing direct observation of patient anatomy and the surgical field. This enhances the surgeon’s view of the physical environment without any need for invasive preoperative placement of fixed fiducial markers. Augmented reality offers real-time image registration without the need for markers [12].

This study involved two steps: 

1. Preclinical evaluation of cadaveric anatomical flaps;

2. Clinical evaluation of flap harvesting prior to oral soft tissue reconstruction. Augmented reality was used to guide the anatomical flap design.

## 2. Experimental Section

### 2.1. Preclinical Cadaveric Anatomical Flap Evaluation 

A cadaveric anatomical assessment of the vascular patterns of the superficial temporal vessels was performed. We dissected four fresh (two male, two female) and four fixed (three male, one female) cadavers obtained from the Body Programme of Bologna University, Bologna, Italy, between June 2018 and October 2019. Working in the “Giovanni Mazzotti” anatomical dissecting room of the Institute of Anatomy, University of Bologna, we harvested flaps and assessed the anatomy and vascular patterns on both sides of each head. Cadaver age, race, height, weight, parity, and cause of death were recorded. We used a technical-grade embalming solution (PanReac Applichem, Barcelona, Spain).

### 2.2. Cadaver Colouration and Fixation

To expose the superficial temporalis artery (STA) and superficial temporalis vein (STV) and study the micro-vascularisation, red and blue latex solutions, respectively, were injected into seven of the eight cadavers. We qualitatively assessed the extent of arterial and venous filling. The STA and STV were identified in each preauricular region after dissection. The skin and subcutaneous tissue were removed and assessed in terms of injection fluid penetration. Each vessel was isolated 1–1.5 cm from surrounding tissue and cannulated with a curved blunt needle inserted 2–3 cm into the vessel, and secured with 6–0 silk sutures to prevent dislocation and leakage.

### 2.3. Anatomical Dissection

A coronal surgical approach was used to expose the entirety of the frontal, temporal, and parietal regions. It was thus possible to separate the anterior and posterior branches of the STA and STV, as shown in Figure 1.

To preserve the superficial temporal vessels, a cutaneous flap was raised along the level of the subcutaneous tissue. A wide temporoparietal flap was supplied by the superficial temporal circulation. At this point, flap harvesting could be performed in a cranial–caudal direction, that is, along the path of the vessel layer, until the temporal fascia (which was included in the flap) was reached. The position of the STA dictated the dissection depth. The pericranium was peripherally incised after the relevant area had been marked. The pericranium was dissected using a periosteal freer, with great care being taken to avoid tearing. This completed the flap harvesting. 

Dissection proceeded in a subfascial manner until the temporal line was reached. The temporal line served as the anatomical landmark indicating the end of the fascia, fascial insertion into temporal bone, and spreading of the pericranial layer. The flap was detached, and the maximum transverse diameter was measured. The calibre of the STA and STV was also assessed, as shown in Figure 2.

### 2.4. Clinical Study

The study was approved by an ethical committee (217/2017/O/sper). In an in vivo study, we used our cadaveric experience to harvest flaps from eight patients. The patients had undergone ablative surgery to treat oral squamous cell carcinoma (OSCC) and were candidates for maxillary/upper alveolar gingiva reconstruction using a composite galea-pericranium pedicled flap, as described below. Since this is designed as a viability study without previous clinical robust data demonstrating the efficacy of bone regeneration from a galeo-pericranial flap, we decided to apply this reconstruction to anatomical defects where bone free flaps were considered as an overtreatment. Therefore, we identified small-size composite maxillary defects (involving both soft and hard tissues) as the ideal target of our research. Mandibular defects were avoided due to the anatomical distance from the donor site.

## 3. Anatomical Flap Design Under AR Guidance was Part of the Study Protocol

### 3.1. Preparation for AR Guidance

Each patient underwent computed tomography of the head and neck region. The vascular pattern of the superficial temporal vessels was assessed, and vascular segmentation and three-dimensional (3D) modelling were performed at the Laboratory of Bioengineering of the University of Bologna, in cooperation with maxillofacial surgeons. We used D2P^TM^ software (DICOM to Print; 3D Systems Inc., Rock Hill, SC, USA) to convert DICOM medical images into 3D digital models for surgical planning (Appendix A). Each STereo Lithography format file (STL file) was imported into Unity 3D software (Unity Technologies, San Francisco, CA, USA). Registration between virtual content (i.e., the 3D model of the temporal vessels) and real anatomy was performed using the Model Target function of Unity 3D. The ear served as an anatomical landmark during registration. Holographic vascular overlays were generated and superimposed on the real anatomy. The surgeon could access the results on a tablet computer. After registration was completed and the correspondence between the virtual and real surface anatomy verified, the entire path of the temporal artery was marked on the skin under AR guidance as shown in the Appendix A.

### 3.2. Surgical Procedure

The surgical access route was the same as for harvesting a muscle rotation flap, that is, the incision extended from the preauricular region to the upper temporoparietal skin. After incising the skin in the preauricular region, the main trunks of the STA and STV were separated, and the skin flap was elevated by sharp dissection along the subcutaneous layer to preserve the superficial temporalis blood supply. After achieving the extended surgical exposure required to raise a galea-pericranium flap, the pericranium was incised and detached from the bone, with great care being taken to preserve tissue integrity, as shown in Figure 3.

Flap harvesting continued in a cranial–caudal direction; the deep temporal fascia remained attached to the temporal muscle, as shown in Figure 4.

It was now possible to totally isolate the vessels of the preauricular region. This allowed us to complete harvesting and mobilisation of the deep portion of the galea-pericranium flap, as shown in Figure 5.

The flap was then tunnelled into and rotated within the oral cavity (below the zygomatic arch) to reconstruct the surgical defect, as shown in Figure 6.

One of the eight flaps included parietal skin, as shown in Figure 7.

## 4. Results

### 4.1. Cadaver Harvesting

During the cadaveric study, it was possible to isolate and follow the entire paths of the bilateral STA and STV. Sixteen flaps were prepared. The anatomical blood supply to the pericranial parieto-temporal region appeared to arise from the superficial temporal vessels and reached the cranial bone via perforator vessels, as shown in Figure 8.

The maximum width of the temporoparietal flap was 15 cm; the appropriate width for each patient depended on the vascular pattern of the STA. The average vessel diameter was 4 mm and the average pericranial flap was 7 cm × 12 cm (range: 6–8 cm × 10–13 cm). When required, the lateral extension provides adequate soft tissue bulk for oral reconstruction. The range of flap rotation was also tested; the flap easily reached the maxillary, cheek, and ipsilateral mandibular regions.

### 4.2. Clinical Outcomes

Eight patients were treated between January 2019 and February 2020 at the Oral and Maxillofacial Surgery Unit, at IRCCS Policlinico Sant’Orsola, Bologna, Italy. Five males and three females with an average age of 73.8 years (range: 65–82 years) were recruited for the study. All of the patients had been diagnosed with OSCC involving the upper alveolar gingiva or the hard palate, and underwent surgical reconstruction using galea-pericranium pedicled flaps. Three patients required reconstruction of hard palate defects, four required reconstruction of the upper alveolar gingival and cheek region, and one required reconstruction of both soft and hard palate defect. Table 1 summarizes the patients’ characteristics.

The average follow-up was nine months (range: 2–15 months). All of the reconstructions were successful and there were no major postoperative vascular complications. All flaps completely restored the oral cavity anatomy in terms of both tissue quality and quantity. No minor postoperative complications (e.g., partial flap loss caused by dehiscence/oroantral communication) were noted. Temporary paralysis of the frontal branch of the facial nerve was observed in one case. No postoperative functional issues at either the donor or recipient site were encountered; all patients healed well, as shown in Figure 9 and Figure 10.

The AR guidance system accurately represented the anatomical vascular patterns of the temporal vessels and subjectively was helpful when designing the flaps.

## 5. Discussion

Oral surgical reconstructive procedures have always been both fascinating and challenging for reconstructive surgeons. Extensive complex defects involving the mucosa, muscle, and bone often require radical approaches, including the use of composite revascularized flaps [2,3]. Small-to-medium defects are best covered with local flaps, such as temporalis muscle or temporalis fascia flaps. Galea and temporalis fascia flaps are usually reserved for head and neck reconstruction [1,4,7,8,9].

Although some anatomical studies assessed the layer structure and arterial anatomy of the temporal region in the context of the vascular basis of various temporal flaps [9], no study has explored the viability of a temporoparietal/periosteum composite flap for oral defect reconstruction. We suspected that the cranial region would provide surgeons with more options, in terms of harvesting periosteum flaps to reconstruct soft and hard tissue defects.

Some researchers have recently used periosteum flaps for bone replacement and regeneration. In a technical note, Bettoni et al. [4] described the use of a radial periosteum free flap to treat early jaw osteoradionecrosis; the bone regeneration outcome was good. Medial femoral condyle, osteoperiosteum and periosteum-only free flaps have been widely used for bone reconstructions requiring minimal extension, such as nasal bone reconstructions; the bone regeneration outcome was also good [5,6]. However, flap harvesting is difficult, and the periosteum is poor in size. Thus, further research is needed.

In this study, we first verified that the superficial temporalis vessels supplied blood not only to the galea and fascia above the temporalis muscle, but also to the temporoparietal and pericranial regions. Our cadaveric study revealed consistent superficial temporalis vascularisation, involving anterior and posterior branches of the STA and STV. Moreover, we found that perforating vessels always ran directly from the STA deep into the pericranium. When we compared radial and medial femoral condyle periosteum flaps with the cadaver-harvested galea-pericranium flap, the amount of periosteum that could be harvested was much greater for the latter flap. The average periosteum area of the galea-pericranium flaps was 7 cm × 12 cm, which was almost double that harvested with radial and medial femoral condyle flaps. Additionally, the average diameter of the temporalis vessels (4 mm) was appropriate for eventual anastomosis.

The blood supply pattern and the cadaveric findings encouraged us to investigate whether a pedicled galea-pericranium flap could be harvested in vivo. In the clinical phase of our study, we used the new flaps to reconstruct oral cavity defects. Although flap harvesting is not easy, the flaps were viable after harvesting and after insertion into and rotation within the oral cavity. Clinical outcomes were good, and no soft tissue retraction occurred. Complete mucosal covering was evident after 2 months.

Our present stage of research encourages us to prosecute along this path in the next phase of our study which will assess the possible osteogenic role of the above-described flap in order to expand its potential clinical applicability for bone regeneration. In fact, follow-up CT scans will be used to identify the presence of radiological features confirming bone formation in the surgical fields.

AR-assisted navigation during flap harvesting allowed us to identify the flap design requirements based on patient anatomy. Moreover, the risk of damage to the vascular temporal pedicle was minimised by separating vessels prior to incision.

Traditionally, the same aim can be achieved using doppler ultrasound to mark out the vessels.

However, the AR significantly improves both the planning and harvesting of the flap procedure. The AR technology merges the anatomy of the patient with the images of the virtual planning (dimension of the flap in accordance with the vascular pattern, anatomical structures of interest, distance from the defect site), thereby representing an enhanced scene for the surgeon’s eye. Augmented reality can also display, in a single scene, additional information allowing the surgeon to plan the best flap design. In lieu of classical doppler ultrasound, this scenario can be obtained with a 3D visualization method.

This is of course of great help in terms of a fully 3D surgical field visualization before and during surgery.

Very few studies have used AR when harvesting reconstructive flaps. Augmented reality greatly assists craniomaxillofacial surgery, overcoming the need for soft tissue reference marks and registration. In a previous study [12], we described our AR-assisted navigation system and computer-aided design/computer-aided manufacturing (CAD/CAM) method of mandibular reconstruction. Here, we confirmed the robustness of this technique, which can be used to determine the accessibility of the superficial anatomy, including the temporal region, and the associated vascular patterns.

## 6. Conclusions

We found that a new type of periosteum flap (galea-pericranium flap) could be harvested from the temporoparietal region. The flap is morphologically and functionally reliable, as revealed by both a preclinical cadaveric study and successful flap placement in eight patients. Augmented reality is used to facilitate the design of flaps that can be employed for oral soft tissue reconstruction, including in cases with medium-sized defects. The promising results have encouraged us to investigate whether the flap is osteogenic, which would expand its clinical utility.

## Figures and Tables

**Figure 1 jcm-09-02211-f001:**
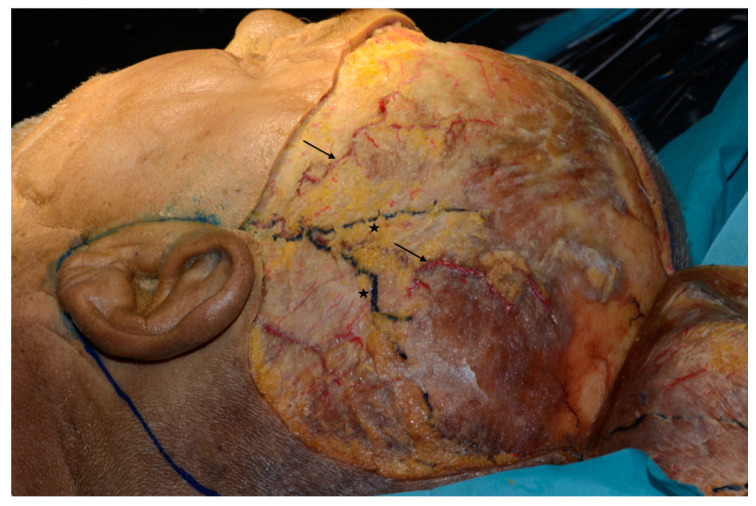
Exposure of the entirety of the frontal, temporal, and parietal regions of fixed cadavers. The superficial temporal artery, anterior and posterior branches (arrows) and superficial temporal vein, anterior and posterior branches (stars) are shown.

**Figure 2 jcm-09-02211-f002:**
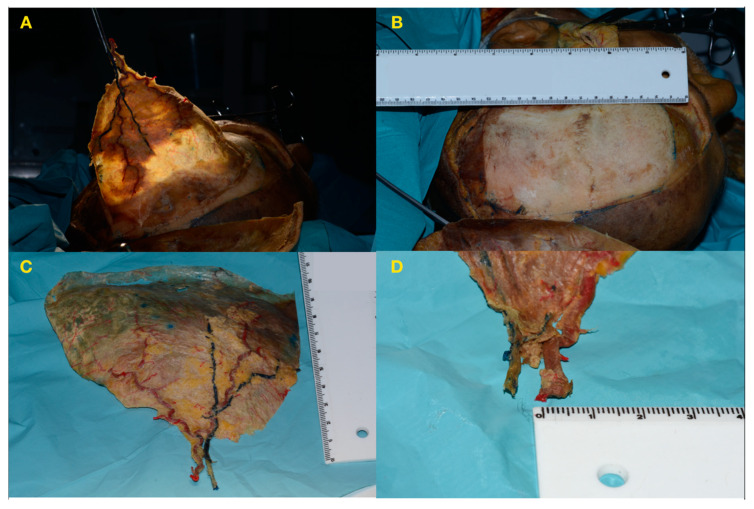
Completion of pericranium dissection and flap harvesting. Detachment and measurement of the flap and vessels. (**A**): A large view. (**B**): Donor area measurement. (**C**): Flap measurement. (**D**): Vessel measurement.

**Figure 3 jcm-09-02211-f003:**
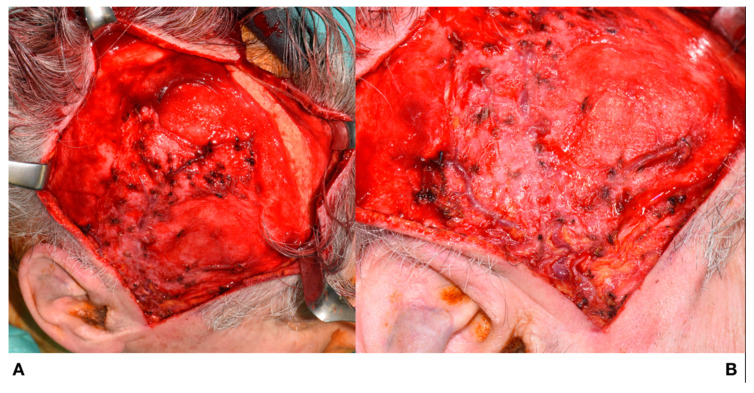
(**A**): The pericranium is incised followed by blunt dissection of the pericranium from the bone. (**B**): Skin flap showing preserved superficial temporalis vessels.

**Figure 4 jcm-09-02211-f004:**
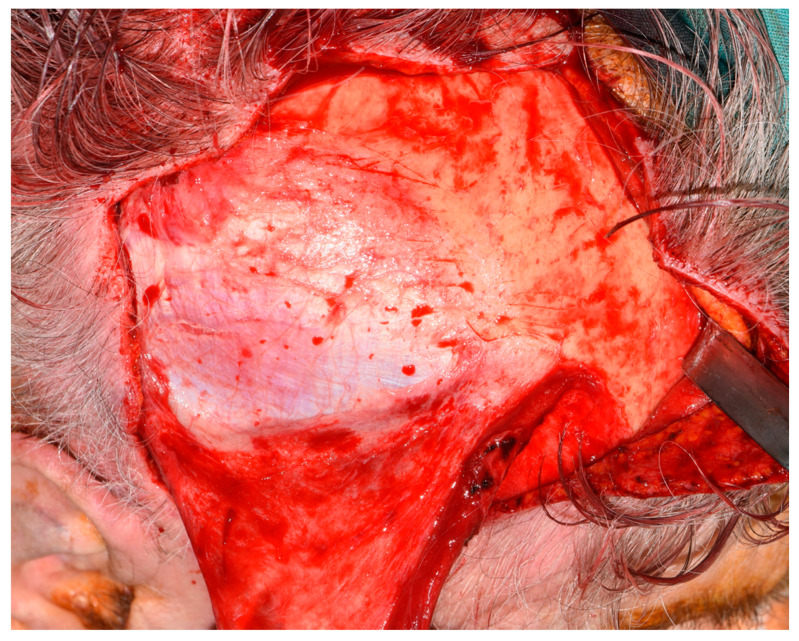
Flap harvesting. The deep temporal fascia remains attached to the temporal muscle.

**Figure 5 jcm-09-02211-f005:**
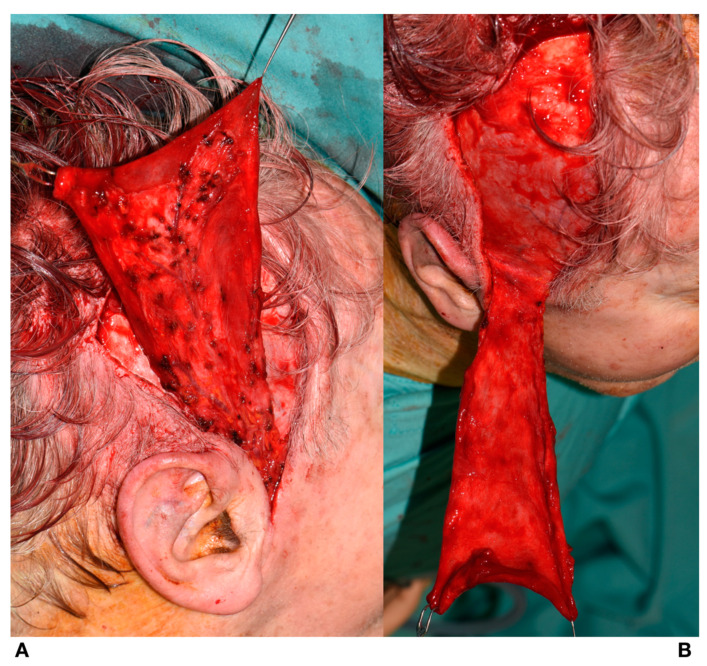
Raising of the flap. (**A**): Superficial view. (**B**): Deep view.

**Figure 6 jcm-09-02211-f006:**
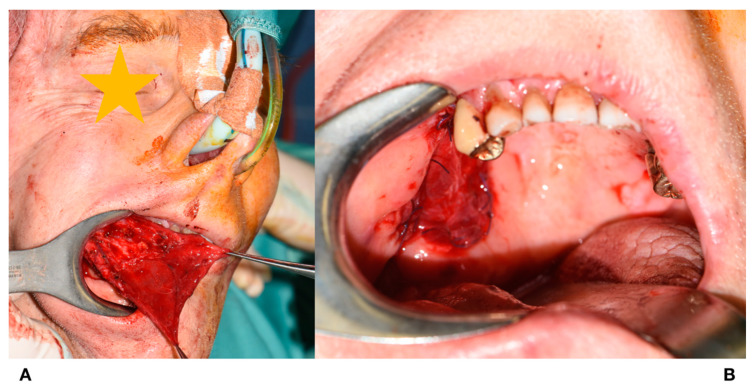
Tunnelling and rotation of the flap. (**A**): Flap rotation in the oral cavity. (**B**): Insertion of the flap.

**Figure 7 jcm-09-02211-f007:**
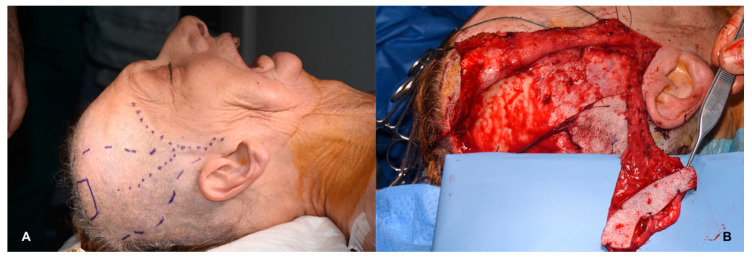
Comprehensive harvesting of a composite skin flap. (**A**): Augmented reality (AR)-assisted flap design prior to skin incision. (**B**): Raising of the flap.

**Figure 8 jcm-09-02211-f008:**
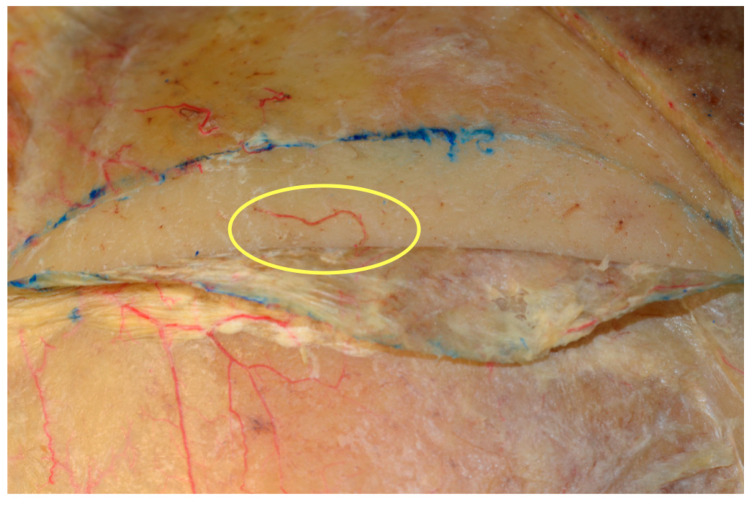
The anatomical blood supply to the parieto-temporal pericranial region. Perforators arising from superficial temporal vessels are visible in both embalmed and fresh cadavers.

**Figure 9 jcm-09-02211-f009:**
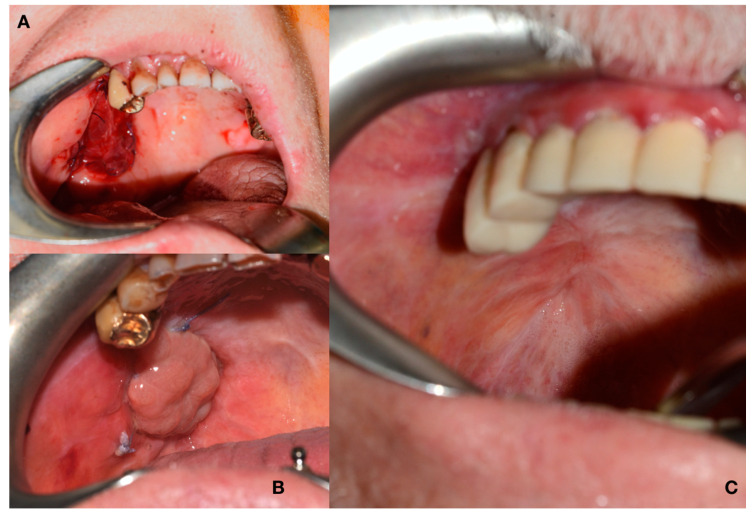
Reconstruction outcomes. Case 1. (**A**): The intraoral insert. (**B**): 2 weeks after surgery. (**C**): 3 months after surgery.

**Figure 10 jcm-09-02211-f010:**
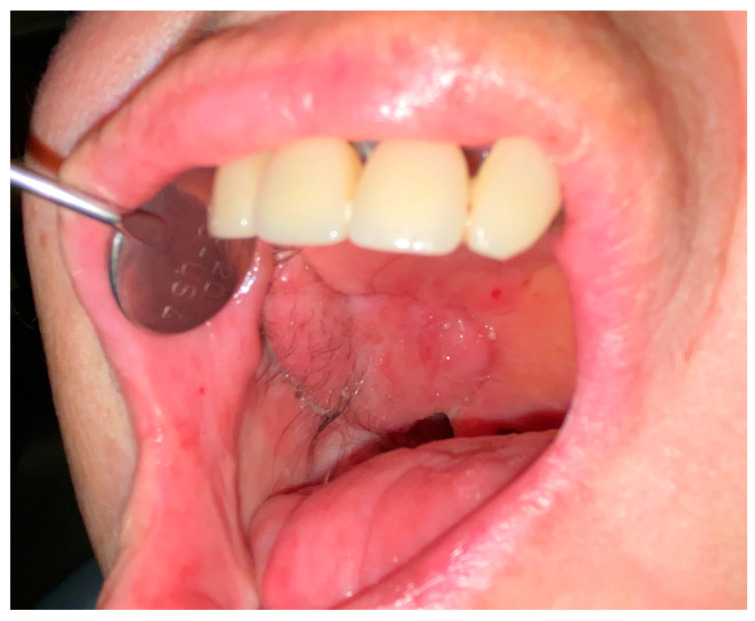
Further reconstruction outcomes. Case 2 at 2 months after surgery. Right palate healing after composite skin/pericranium flap transfer.

**Table 1 jcm-09-02211-t001:** Demographic and clinical characteristics of enrolled patients.

Patient	Sex	Pathology	Site of Defect	Flap	Complications
**#1**	M	OSCC	Upper alveolar gingiva	Galeo-pericranial flap	-
**#2**	M	OSCC	Upper alveolar gingiva/cheek	Galeo-pericranial flap	-
**#3**	F	OSCC	Upper alveolar gingiva	Galeo-pericranial flap	-
**#4**	M	OSCC	Upper alveolar gingiva	Galeo-pericranial flap	-
**#5**	M	OSCC	Hard palate	Galeo-pericranial flap	-
**#6**	F	OSCC	Hard palate	Galeo-pericranial flap	-
**#7**	M	OSCC	Hard palate	Galeo-pericranial flap	-
**#8**	F	OSCC	Hard and soft palate	Galeo-pericranial flap including the parietal skin	Temporary paralysis of the frontal branch of the facial nerve

OSCC: oral squamous cell carcinoma.

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
