# Peer review of "Augmented Reality-Assisted Periosteum Pedicled Flap Harvesting for Head and Neck Reconstruction: An Anatomical and Clinical Viability Study of a Galeo-Pericranial Flap"

_jcm, 2020, doi:10.3390/jcm9072211_

Round 1

Reviewer 1 Report

This is an interesting report of use of augmented reality for flap planning and use of the galeo-pericranial flap in head and neck reconstruction. 

  1. It is unclear what you envision as the use of this flap?  Do you see this flap being used for segmental mandible defects and/or maxillary reconstruction?  Please clarify this in the manuscript. 
  2. What were your expected outcomes for the patients presented in the paper?  Were you expecting to see new bone growth?  Are you planning for future scans to assess new bone?  Please clarify this in the manuscript.
  3. What is the advantage of AR over just using Doppler ultrasound to mark out the vessels?  Please clarify this in the manuscript.
  4. A table summarizing the patient demographics, defect location, complications, etc would be helpful considering there are only 8 patients to report.

Author Response

Dear Reviewer,

We thank you for your interest in our work and for the time you spent reviewing it.

We have carefully considered the comments provided and we have revised the paper accordingly.

Point 1:It is unclear what you envision as the use of this flap?  Do you see this flap being used for segmental mandible defects and/or maxillary reconstruction?  Please clarify this in the manuscript. 

Response 1: Since this is designed as a viability study without previous clinical robust data demonstrating the efficacy of bone regeneration from galeo-pericranial flap, we decided to apply this reconstruction to anatomical defects (including the the bone) where bone free flaps were considered as an overtreatment. So, we identified small size maxillary defects (involving both soft and hard tissues) as the ideal target of our research. Mandibular defects were avoided due to the anatomical distance from the donor site. This specification has been added in the 'Clinical Study' section. In addition, we have added a table describing the site of the defect for each enrolled patient.   Point 2: What were your expected outcomes for the patients presented in the paper?  Were you expecting to see new bone growth?  Are you planning for future scans to assess new bone?  Please clarify this in the manuscript.   Response 2: Our present stage of research encourages us to prosecute along this path in the following phase of our study which will lead us to assess the possible osteogenic role of the above described flap in order to expand its potential clinical applicability for bone regeneration. In fact, follow-up CT scans will be assessed in order to identify the presence of radiological features confirming bone formation in the surgical fields. Our preliminary data from CT showed features of bone regeneration. However we have to prospectively assess this aspect to confirm this hypothesis. We have added this aspect in the discussion section.   Point 3: What is the advantage of AR over just using Doppler ultrasound to mark out the vessels?  Please clarify this in the manuscript.   Response 3: Traditionally, the same aim can be reached using doppler ultrasound to mark out the vessels. However, the AR significantly improves both the planning and the harvesting flap way. This technology merges the anatomy of the patient with the images of the virtual planning (dimension of the flap in accordance to the vascular pattern, anatomical structures of interest, distance from the defect site), representing in this way an enhanced scene for the surgeon’s eye. AR can also display in a single scene additional information allowing the surgeon to plan the best flap design. Despite of classical doppler ultrasound, this scenario can be obtained with a 3D visualization method. We have discussed this issue in the paper.   Point 4: A table summarizing the patient demographics, defect location, complications, etc would be helpful considering there are only 8 patients to report.   Response 4: Thank you for your suggestion. We have introduced a table describing the study population.

Reviewer 2 Report

Aim of this prospective study is to determine whether a new type of periosteum flap, the galeo-pericranial flap with superficial temporalis vascularisation can be harvested and used as bone graft substitute with the help of augmented reality. The study involved two steps: Preclinical evaluation of cadaveric anatomical flaps and Clinical evaluation of flap harvesting prior to oral soft tissue reconstruction. AR was used to 62 guide the anatomical flap design.

The article is interesting and well written, with clear structure and good English. Figures are informative but could be less. The video is very useful. References are current but could be improved.

Fig could be reduced. Fig.2 could be removed

Furthermore I would like to make following Comments:

Was there any evidence of bone regeneration in your study group?

190 “The AR guidance system accurately represented the anatomical vascular patterns of the temporal vessels…… and subjectively was helpfull when designing the flaps.” Please correct:  Since you did not perform any surgery without AR you can not have an objective comparison about how helpful AR really was.

Please discuss alternatives to AR for harvesting this flap: e.g. using color doppler sonography would probably have the exact same results

Author Response

Dear Reviewer,

We thank you for your interest in our work and for the time you spent reviewing it.

We have carefully considered the comments provided and have revised the paper accordingly.

Point1: The article is interesting and well written, with clear structure and good English. Figures are informative but could be less. The video is very useful. 

Fig could be reduced. Fig.2 could be removed

Response: Figure 2 has been removed.

Point 2: Was there any evidence of bone regeneration in your study group?

Response 2: Since this is designed as a viability study without previous clinical robust data demonstrating the possibility of harvesting a galeo-pericranial flap for oral reconstruction, we studied the vascular supply to the pericranial temporoparietal region to determine whether a galeo-pericranial flap with superficial temporalis vascularisation could be harvested.

Bone regeneration was not included as endpoint of the present study. However, the present stage of research encourages us to prosecute along this path in the following phase of our study which will lead us to assess the possible osteogenic role of the above described flap in order to expand its potential clinical applicability for bone regeneration. In fact, follow-up CT scans will be assessed in order to identify the presence of radiological features confirming bone formation in the surgical fields. From our preliminary radiographic findings we identified bone regeneration in the surgical site. We have to prospectively assess CT-data to confirm this hypothesis.

We will work on a further paper focusing on this topic.

We have added this aspect in the discussion section.

Point 3: “The AR guidance system accurately represented the anatomical vascular patterns of the temporal vessels…… and subjectively was helpfull when designing the flaps.” Please correct:  Since you did not perform any surgery without AR you can not have an objective comparison about how helpful AR really was.

Response 3: Thank you for your suggestion. We have correct the statement accordingly. 

Point 4: Please discuss alternatives to AR for harvesting this flap: e.g. using color doppler sonography would probably have the exact same results.

Response 4: Traditionally, the same result can be reached using doppler ultrasound to mark out the vessels.

However, the AR significantly improves both the planning and the harvesting flap way. This technology merges the anatomy of the patient with the images of the virtual planning (dimension of the flap in accordance to the vascular pattern, anatomical structures of interest, distance from the defect site), representing in this way an enhanced scene for the surgeon’s eye. AR can also display in a single scene additional information allowing the surgeon to plan the best flap design. Despite of classical doppler ultrasound, this scenario can be obtained with a 3D visualization method.

This is of course is of great help in terms of a fully 3D surgical field visualization before and during surgery. 

We have introduced this aspect in the discussion section.

We would like to thank you for your revision because we think that your suggestions have improved the quality of this paper.